# Connecting Data to Mechanisms with Meta Structural Causal Model

## Abstract

Recent years have seen impressive progress in theoretical and algorithmic developments of causal inference across various disciplines in science and engineering. However, there are still some unresolved theoretical problems, especially for cyclic causal relationships. In this article, we propose a meta structure causal model(meta-SCM) framework inspired by understanding causality as information transfer. A key feature of our framework is the introduction of the concept of *active mechanisms* to connect data and the collection of underlying causal mechanisms. We show that the meta-SCM provides a novel approach to address the theoretical complications for modeling cyclic causal relations. In addition, we propose a *sufficient activated mechanisms* assumption, and explain its relationship with existing hypotheses in causal inference and learning. Finally, we conclude the main idea of the meta-SCM framework with an emphasis on its theoretical and conceptual novelty.

## 1 Introduction

Although there have been significant advances in causal research in recent years, there are still some important theoretical problems that have not been resolved. One of the most notoriously hard problems is about cyclic causal relations, and there is no causal modeling frameworks can properly handle it. In modern theory of causality, the mathematical framework called a structural causal model (SCM) is used to represent the causal mechanisms from which a causal hierarchy to describe the generated phenomena organically emerges(Pearl, 1995; Pearl et al., 2009; Bongers et al., 2016; Bareinboim et al., 2020). Acyclic SCMs, also known as recursive SEMs, form a special well-studied subclass of SCMs that generalize causal Bayesian networks. They have many convenient properties and are widely used in practical causal modeling, see e.g. (Evans, 2016; Lauritzen, 1996; Richardson, 2003; Maathuis et al., 2018). But there is a strong need to go beyond acyclic SCMs.

In fact, there are feedback loops between observed variables in many systems occurring in real world. Causal cycles may arise when one approximates such systems over time (Fisher, 1970; Mogensen et al., 2018; 2020), or when one describes the equilibrium states of these systems (Iwasaki & Simon, 1994; Lacerda et al., 2012; Hyttinen et al., 2012; Mooij et al., 2013; Bongers & Mooij, 2018; Blom et al., 2020; Pfister et al., 2019), though the underlying dynamic processes describing such systems have an acyclic causal structure over time. In particular, it was shown that the equilibrium states of a system governed by (random) differential equations can be described by an SCM that represents their causal semantics in (Bongers & Mooij, 2018), which gives rise to a plethora of SCMs that include cycles. In contrast to their acyclic counterparts, many of the convenient properties do not hold for SCMs with cycles, and they are not as well understood. Some progress has been made in the case of discrete (Neal, 2000) and linear models (Spirtes, 1993; 1994; 2013; Richardson et al., 1996; Koster et al., 1996; Hyttinen et al., 2012), and more recently the Markov properties (Forré & Mooij, 2020; 2017) and theoretical foundation (Bongers et al., 2016). Researchers are mostly making additional assumption of the underlying causal mechanisms to circumvent complications of cyclic SCMs in causal semantics, solvability, marginaliztions etc.. However, they are still not well understood. Even, a pressing concern is whether SCMs are able to completely model dynamical systems at equilibrium and the causal constraints model(CCM) is proposed but without graphical interpretations yet(Blom et al., 2020).

After introducing the formal definition of SCMs and relevant preliminaries in Section 2, we trace back into the philosophy accounts of causality and propose the meta structural causal model (meta-SCM) based on understanding causality as information transfer in Section 3. The meta-SCM framework is constructed by an SCM and an extra dimension which describes how to connect data to mechanisms through the concept of *active mechanisms*. In particular, a meta-SCM induces a sub-model for each sample in the dataset coarsely related to the active set. The new framework is proved to be more expressive by Theorem 7, and its potential ability to address cyclic casual relationships is illustrated by an example of cyclic SCM with multiple solution and an unsolvable cyclic SCM. Comparing to the joint causal inference for meta-system (Mooij et al., 2016) which can also deal with cyclic SCMs, the meta-SCM framework avoids to add extra context variables and can even gain insights on unsolvable SCMs.

The Section 4 addresses the challenging case where no additional variables, besides the samples from the data to generate, are observed. The sufficient activated mechanism (SAM) hypothesis is proposed as an central assumption in the meta-SCM framework, which is consistent with the role of independent causal mechanisms (ICM) principle or sparse mechanisms shift (SMS) assumption for the SCM framework. Moreover, the SAM and SMS hypotheses are also compared with the lens of informational decomposition of SCM, and it reveals that the SMS assumption might be not appropriate in certain case with an example. Section 5, we conclude the main idea of the meta-SCM framework with an emphasis on its theoretical and conceptual novelty.

The main contributions of this paper are:

1) We propose a totally novel dimension that describes how to link data to mechanisms to the existing causal modeling framework, particularly, a meta structural causal model framework which can be used to circumvent technique complications in cyclic SCMs.

2) We propose a SAM hypothesis as an inductive bias for performing causal inferences and learning consistent with the role of SMS assumption.

## 2 PRELIMINARIES

At the centre of modern causal modeling theory lies the structural causal model (SCM) (also known as structural equation model) which makes graphical assumptions of the underlying data generating process. There are many somewhat different formulations of SCM in literatures, e.g., Schölkopf (2019); Pearl (2019); Bongers et al. (2016); Pearl et al. (2009); Forré & Mooij (2020), among which the definition in Blom et al. (2020) is used in this paper.

**Definition 1 (SCM)** *Let $\mathcal{I}$ and $\mathcal{J}$ be index sets. A Structural Causal Model (SCM) $\mathcal{M}$ is a triple $(\mathcal{X}, F, \mathbf{E})$, with:*

- *a product of standard measurable spaces $\mathcal{X} = \Pi_{i \in \mathcal{I}} \mathcal{X}_i$ (domains of endogenous),*

- *a tuple of exogenous random variables $\mathbf{E} = (E_j)_{j \in \mathcal{J}}$ taking value in a product of standard measurable space $\mathcal{E} = \prod_{j \in \mathcal{J}} \mathcal{E}_j$.*

- *a family of $F$ of measurable functions:*

$$f_i : \mathcal{X}_{pa(i) \cap \mathcal{I}} \times \mathcal{E}_{pa(i) \cap \mathcal{J}} \to \mathcal{X}_i, \quad \forall i \in \mathcal{I}.$$

The dataset are ususally assumed to be a set of samples for the solution of SCMs.

**Definition 2 (Solution of SCM)** *We say that a random variable $X = (X_i)_{i \in \mathcal{I}}$ is a solution to an SCM $\mathcal{M} = (\mathcal{X}, F, \mathbf{E})$ if*

$$X_i = f_i(\mathbf{X}_{pa(i) \cap \mathcal{I}}, \mathbf{E}_{pa(i) \cap \mathcal{J}}) \quad a.s., \quad \forall i \in \mathcal{I}$$

*An SCM may have a unique(up to zero sets) solution, multiple solutions, or there may not exist any solution at all.*

**Definition 3** *An SCM $\mathcal{M}$ is called* simple *if it is uniquely solvable with respect to any subset $\mathcal{O} \subseteq \mathcal{I}$.*

All acyclic SCMs are simple.

**Definition 4** *A do intervention $do(\tilde{x}_I)$ with target $I \subseteq \mathcal{I}$ and value $\tilde{x}_I \in \mathcal{X}_I$ on an SCM $\mathcal{M} = (\mathcal{X}, F, \mathbf{E})$ maps it to the intervened SCM $\mathcal{M}_{do(\tilde{x}_I)} = (\mathcal{X}, \tilde{F}, \mathbf{E})$ with $\tilde{F}$ the family of measurable functions:*

$$\tilde{f}_i(\mathbf{x}_{pa(i) \cap \mathcal{I}}, \mathbf{e}_{pa(i) \cap \mathcal{J}}) = \begin{cases} \tilde{x}_i & i \in I, \\ f_i(\mathbf{x}_{pa(i) \cap \mathcal{I}}, \mathbf{e}_{pa(i) \cap \mathcal{J}}) & i \in \mathcal{I} \setminus I \end{cases}$$

The intervened SCM is referred as a submodel of the original SCM, in fact, the variants derived from many different types of interventions (e.g., perfect, imperfect, stochastic, etc.) are also referred as submodel.

## 3 META STRUCTURAL CAUSAL MODELS

One critical insight in philosophy is that the causal mechanisms behind a system under investigation are not generally observable, but they do produce observable traces ("data," in modern terminology). This insight naturally leads to two practical desiderata for any proper framework for causal inference, namely:

1. The causal mechanisms underlying the phenomenon under investigation should be accounted for – indeed, formalized – in the analysis.
2. This collection of mechanisms (even if mostly unobservable) should be formally tied to its output: the generated phenomena and corresponding datasets.

The mathematical object called a structural causal model (SCM) is used to represent the causal mechanisms from which a causal hierarchy to describe the generated phenomena organically emerges. It is often assumed that every instantiation $\mathbf{E} = \mathbf{e}$ of the exogenous variables uniquely determines the values of all variables in $\mathbf{X}$ (Pearl, 2019), which leads a unique solution of the corresponding SCM . Then the dataset $\mathcal{D} = \{\mathbf{x}^{(k)}\}_{k=1,\ldots,N}$ is a set of $N$ samples of the unique solution. But in many cases, SCM with cycles might be not solvable or have multiple solutions (Halpern, 1998).

**Example 1 (Multiple Solutions)** *Consider an SCM $\mathcal{M}^1 = (\mathcal{X}, F = \{f_1, f_2\}, \mathbf{E} = \{E_1\})$, where*

$$F = \begin{cases} x_1 \leftarrow (x_2^2 + x_2 + 1)/3 - e_1^2/3 \\ x_2 \leftarrow x_1 \end{cases}$$

*Obviously, $(1 - E_1, 1 - E_1)$ and $(1 + E_1, 1 + E_1)$ are two different solutions to $\mathcal{M}^1$, then which solution of the SCM should be used to link the dataset to the model?*

The previous causal inference literature rarely deals with theoretical aspects of cyclic causality. In recent years, it has been formally discussed in Bongers et al. (2016) However, it is also acknowledged by this paper that there are many complications in dealing with cyclic causal models. The vast majority of methods to deal with cyclic SCM in the literature are by adding additional assumptions, such as linear constraints(Spirtes, 1993; 1994; Hyttinen et al., 2012) and certain solvability constraints(Forré & Mooij, 2018; Bongers et al., 2016). These methods basically exclude the study of SCMs with multiple solutions such as Example 1.

The view of understanding causation as information transfer was first formally proposed by (Collier, 1999) in philosophy recently. Inspired by this view, we realize that the causal links among variables can be cut off suggested by unsuccessful information transfer, which suggests that different samples might have different causal graphs and causal mechanisms. For example, samples of $\mathcal{M}^1$ might only satisfy only a subset of the structural equations due to absence of information , hence variables for two different samples $\mathbf{x}^{(i)}$ and $\mathbf{x}^{(j)}$ could have two different causal graphs. However, we usually do not know when and where the information transmission was interrupted for a given sample. In fact, it might be infeasible to specify the information transmission details of all samples of an SCM. To address the above problem, we introduce the concept of *active set* of an SCM, which is inspired by the *active set method* in the field of non-linear optimization theory.

**Definition 5 (Active mechanisms)** *For a given sample $\mathbf{x}^{(k)}$ and the corresponding collection of mechanisms represented by an SCM $\mathcal{M} = (\mathcal{X}, F, \mathbf{E})$, if $x_i^{(k)} = f_i(\mathbf{x}_{pa(i) \cap \mathcal{I}}^{(k)}, \mathbf{e}_{pa(i) \cap \mathcal{J}}^{(k)})$, then we call $f_i$ an active mechanism, and denote the index set for all active mechanisms as the active set $A_k$.*

The collection of active sets $\{A_k\}_{k=1,2,...}$ gives us the opportunity to avoid considering the details of the information transfer between variables, and to describe the relationship between the data and the model relatively concisely. Formally, we define the meta structural causal model (meta-SCM) as follows:

**Definition 6 (meta-SCM)** *A collection of mechanisms described by an SCM $\mathcal{M} = (\mathcal{X}, F, \mathbf{E})$ with a dataset $\mathcal{D} = \{\mathbf{x}^{(k)}\}_{k=1,...,N}$, in which each sample $\mathbf{x}^{(k)}$ satisfies that:*

- *the prior distribution of $\mathbf{E}^{(k)}$ is $P(\mathbf{E})$;*
- *$A_k \subseteq \mathcal{I}$ is referred as the* active set *of the sample $k$ satisfies that*

$$x_i^{(k)} \leftarrow f_i(\mathbf{x}_{pa(i)\cap\mathcal{I}}^{(k)}, \mathbf{e}_{pa(i)\cap\mathcal{J}}^{(k)}), \quad \forall i \in A_k. \tag{1}$$

*Then the tuple $\langle \mathcal{M}, \mathcal{D} \rangle$(or in short $\mathcal{M}$) is called a* meta structural causal model (meta-SCM)*.*

*The difference between SCM and meta-SCM.* On one hand, an SCM can be interpreted as a special case of meta-SCM satisfies that the active set $A_k$ is equal to $\mathcal{I}$ for any sample $\mathbf{x}^{(k)}$. On the other hand, a meta-SCM share the causal mechanisms with its corresponding SCM only differs on the method for linking data to model. Thus, it improves the expressiveness of the canonical SCM. Actually, a meta-SCM suggests a method for connecting any dataset to an SCM with the active sets. Formally,

**Theorem 7 (Connecting Data to Mechanisms)** *For an SCM $\mathcal{M} = (\mathcal{X}, F, \mathbf{E})$ with any dataset $\mathcal{D} = \{\mathbf{x}^{(k)}\}_{k=1,...,N}$ in the domain of $\mathcal{X}$. Then each datapoint $\mathbf{x}^{(k)}$ is a sample from some submodel SCM $\tilde{\mathcal{M}}$ related to the active set $A_k$.*

**Proof** For any $k = 1, ..., N$, let $\tilde{\mathcal{M}}(k) = (\mathcal{X}, \tilde{F}, \mathbf{E})$ be an SCM with modified causal mechanisms:

$$\tilde{f}_i(\mathbf{x}_{pa(i)\cap\mathcal{I}}, \mathbf{e}_{pa(i)\cap\mathcal{J}}) = \begin{cases} f_i(\mathbf{x}_{pa(i)\cap\mathcal{I}}, \mathbf{e}_{pa(i)\cap\mathcal{J}}) & i \in A_k, \\ x_i^{(k)} & i \in \mathcal{I} \setminus A_k \end{cases}$$

Then the active set of datapoint $\mathbf{x}^{(k)}$ for the submodel $\tilde{\mathcal{M}}(k)$ is $\mathcal{I}$ by definition[1], which directly leads to our theorem. ∎

The above proof directly assigns a submodel for each sample in the dataset, which only part of mechanisms in the original SCM holds. In fact, the submodel in our meta-SCM framework does not have to be constructed as a $\mathrm{do}$-intervened model $\tilde{\mathcal{M}}(k)$, it can be any subclass of SCMs with desired properties(such as acyclic) in literatures.

When there are cyclic causal relationships between variables, one encounters various technical complications, which even arise in the linear setting(Bongers et al., 2016). The main idea for solving related difficulties is to add additional restrictions on structural equations, and the dataset are assumed to be consisted of samples from a distribution obtained by solving the SCM. In contrast, our meta-SCM does not add additional assumptions on SCM, and each datapoint is treated as a sample of the distribution obtained by solving a certain submodel. For the SCM $\mathcal{M}^1$ with muptiple solutions in Example 1 with dataset $\mathcal{D} = \{\mathbf{x}^{(k)}\}_{k=1,...,N}$ in the domain of $\mathcal{X}$, the meta-SCM can circumvent theoretical complications through providing each datapoint a distribution of any solution of a certain submodel. Usually, the details of submodel and its corresponding distribution for each sample might be unknown, and meta-SCM only provides a coarse description by the active sets.

In fact, we can also circumvent the technique complications caused by solvability through meta-SCM. Specifically, the structural equations of an acyclic SCM trivially have a unique solution, which ensures that the SCM gives rise to a unique, well-defined probability distribution on the variables. However, an SCM can be unsolvable in the case of cycles, e.g.,

**Example 2 (Unsolvable)** *Consider an SCM $\mathcal{M}^2 = (\mathcal{X}, F = \{f_1, f_2\}, \mathbf{E} = \{E_1\})$, where*

$$F = \begin{cases} x_1 \leftarrow x_2^2 + x_2 + e_1^2 + 1 \\ x_2 \leftarrow x_1 \end{cases}$$

---

[1]In fact, $\tilde{\mathcal{M}}(k)$ is the do-intervened SCM $\mathcal{M}_{do(\mathbf{x}_{\mathcal{I}\setminus A_k}^{(k)})}$.

Then the SCM $\mathcal{M}^2$ is obvious not solvable, thus it cannot be used to model underlying causal mechanisms of any dataset $\mathcal{D} = \{\mathbf{x}^{(k)}\}_{k=1,...,N}$ in the domain of $\mathcal{X}$. But with our novel approach, we might still connect data to the underlying mechanisms, e.g. by letting $|A_k| = 1$ for all $k = 1,...,N$. In other words, each datapoint $\mathbf{x}^{(k)}$ is a sample of a distribution derived from the submodel $\tilde{\mathcal{M}}(k)$ by Theorem 7.

*The difference between meta-system and meta-SCM.* The joint causal inference (JCI) framework reduces modeling a system in its environment to modeling the meta-system consisting of the system and its environment, which considers auxiliary context variables that describe the context of each data set (Mooij et al., 2016). In contrast, our meta-SCM address the challenging case where no additional variables, besides the samples from the data to generate, are observed. For example in Fig. 1, the meta-system consists of two variables $X_1, X_2$ and a context variable $C$. More concretely, the engine $X_1$ drives the wheels of a car $X_2$ when going uphill $C = 0$, but when going downhill, the rotation of the wheels drives the engine. In a meta-SCM, we instead introduce the concept of active mechanisms to describe each sample in a dataset. Moreover, the meta-SCM framework even gain insights on unsolvable SCMs such as Example 2 while the previous meta-system study restricts themselves to simple SCMs. In philosophy viewpoint, the JCI framework uses context changes to model interventions, which is a difference-making account for causality. Instead, the active set in meta-SCM are inspired by information transferring which is a production account of causality, see e.g. Illari & Russo (2014).

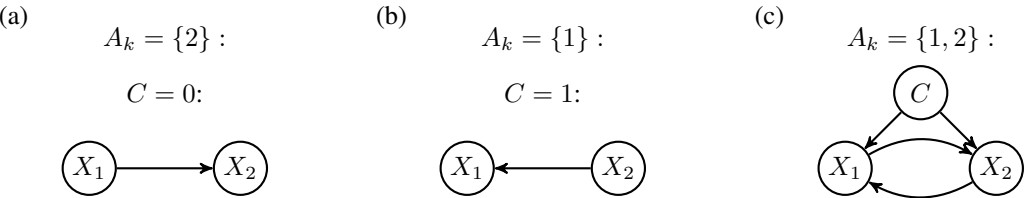

Figure 1: The meta-system and meta-SCM when in the presence of cycles. a) $X_1$ causes $X_2$ in context $C = 0$, $f_2$ is activated; (b) $X_2$ causes $X_1$ in context $C = 1$, $f_1$ is activated; (c) $X_1$ and $X_2$ cause each other in the joint model, $f_1, f_2$ are activated.

From the above discussions, the meta-SCM framework can be considered as generalization of SCM that novelly links data to causal mechanisms through active sets at individual level modeling[2], and it might be used to circumvent technical complications when cycles present. Then, a natural question is how to perform inferences with a meta-SCM when in the presence of cycles? A first difficult might be the lack of knowledge on the specific form of active sets, i.e. $A_k$, for each sample. In the following section, we propose a principle for learning and reasoning within the meta-SCM framework.

## 4 SUFFICIENT ACTIVATED MECHANISMS

The meta-SCM viewpoint considers a set of variables indexed by $\mathcal{I}$ and a set of samples $\mathcal{D}$ with *causal (or disentangled) factorization*,

$$P(\mathbf{x}_{A_k}^{(k)}) = \prod_{i \in A_k} P(x_i^{(k)}|\mathbf{x}_{pa(i)\cap\mathcal{I}}^{(k)}), \quad \forall \mathbf{x}^{(k)} \in \mathcal{D}. \tag{2}$$

The independent causal mechanisms (ICM) principle tells us that the factors should be independent in the sense that

    a) changing (or performing an intervention upon) one mechanism $P(x_i|\mathbf{x}_{pa(i)\cap\mathcal{I}})$ does not change any of the other mechanisms $P(x_j|\mathbf{x}_{pa(j)\cap\mathcal{I}})$ ($i \neq j$) (Schölkopf et al., 2012), and

---

[2]The individual level models is not new but widely used, e.g., in variational inference of machine learning (Bingham et al., 2019).

b) knowing some other mechanisms $P(x_i|\mathbf{x}_{pa(i)\cap\mathcal{I}})$ $(i \neq j)$ does not give us information about a mechanism $P(x_j|\mathbf{x}_{pa(j)\cap\mathcal{I}}))$ (Janzing & Schölkopf, 2010).

Consistent with the implication a) of the ICM Principle, the sparse mechanism shift (SMS) hypothesis (See e.g. Parascandolo et al. (2018); Schölkopf (2019)) was stated in the following:

> **Sparse Mechanism Shift (SMS).** *Small distribution changes tend to manifest themselves in a sparse or local way in the causal/disentangled factorization, i.e., they should usually not affect all factors simultaneously.*

It is the fundamental assumption for causal representation learning(Schölkopf et al., 2021) and has recently been used for learning causal models (Ke et al., 2019), modular architectures (Goyal et al., 2019; Besserve et al., 2020) and disentangled representations (Locatello et al., 2020). However, the SMS hypothesis usually only assumes mechanisms shift from a given context to the original uniquely solvable SCM, especially acyclic SCM. We propose a hypothesis in the viewpoint of a given individual mechanism across samples in the following:

> **Sufficient Activated Mechanism (SAM).** *All factors in the causal/disentangled factorization should be activated sufficiently, i.e., every causal mechanism is sufficiently often activated across samples in the dataset.*

The "sufficient" in the SAM roughly understood as sufficient for identifying factors in the causal factorization of interest. In Example 2, the unsolvable cyclic SCM $\mathcal{M}^2$ cannot be used as the underlying generative process for any dataset. On the contrary, in the meta-SCM framework under SAM assumption, we are able to learn a causal factorization Eq. (2), even when only knowing how many factors have activated, but not which ones. In mathematics, start with solving the unsolvable equation $x^2 = -1$ with imaginary unit $i$, it has been an important branch of mathematics which is complex analysis with holomorphic function as a central research object. Similarly, the meta-SCM, which introduced a novel dimension on linking data to SCM, can help in causal modeling with unsolvable SCM, and we feel meta-SCM with SAM assumption might be a central research object for our framework as holomorphic function for complex analysis.

Overall, in the lens of two practical desirata for any proper framework mentioned in the begining of Section 3, our meta-SCM is different from all literatures on variants of SCMs by adding a fully novel dimension on considering how each sample is tied to underlying mechanisms. This extra dimension though, on one hand, complicate the SCM framework, however, on the other hand, improve its expressiveness and power for causal modeling. Then, the SAM hypothesis defines a subclass of meta-SCMs of interest for performing causal inferences and reasoning, consistent with the role of the ICM Principle/ SMS hypothesis in the SCM framework.

The causality behind interventions in the SCM framework is a difference-making account, while the production accounts, especially informational interpretation of causality relatively neglected. One core idea of informational causality — $C$ causes $E$ if there is information transmission from $C$ to $E$(Illari & Russo, 2014), suggests that the causal mechanism $f_i$ of an SCM can be separated into two part — information process and information transfer (Gong & Zhu, 2021). Formally,

**Definition 8 (Informational Decomposition of SCM)** *Consider an SCM $\mathcal{M} = (\mathcal{X}, F, \mathbf{E})$. The informational decomposition of $\mathcal{M}$ is defined as, for any $i \in \mathcal{I}$,*

$$\begin{cases} x_i \leftarrow f_i(\mathbf{e}_{pa(i)\cap\mathcal{I},i},\ e_{pa(i)\cap\mathcal{J}}), \\ e_{j,i} \leftarrow x_j, \end{cases} \tag{3}$$

*where $e_{j,i}$ represents the information on edge $(j,i)$ received from its input node $j$.*

This dichotomy of causal mechanisms can help illuminate on the relationship between SMS and SAM assumption. We tend to assume the information processing mechanisms are sparse inactivated, while the information transfer mechanisms are instead only sufficient activated. Henceforth, if we consider the inactivated mechanisms as shifted mechanisms, then the information process and

transfer mechanisms are satisfying the SMS hypothesis and the SAM assumption respectively. Small distribution changes caused by interventions on information process tend to manifest themselves in a sparse or local way, but might be manifest themselves in a non-local way caused by changes on transferred information. For example in a starfish shaped causal diagram, changing the output information of the centre node (instead itself) can lead to shift on all functional relationships with its child nodes, which can affect almost all factors simultaneously. Thus, the SMS assumption might be appropriate to describe the intervened model for such system.

## 5    CONCLUSIONS

This work is mostly theoretical and conceptual to address the notoriously hard problem of cyclic causal models. We proposed a novel active set approach for connecting data to SCMs (even not solvable), instead of making additional assumptions to restrict the class of SCMs of interest. Note that the meta-SCM is more of an conceptional modeling framework on how to relate data to the underlying causal mechanisms rather than a specific model, and the SAM assumption has been introduced as an inductive bias for performing causal inferences and learning within the framework of meta-SCM. To best of our knowledge, this is the first causal modeling framework to explore the dimension formally on how to link each sample in the dataset to the collection of mechanisms. In the future, we might anticipate more causal modeling exploration on exacting information from activated mechanisms directly instead of the solution of SCMs.

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
