# OpenReview forum: "Connecting Data to Mechanisms with Meta Structual Causal Model"
_ICLR.cc/2022/Conference — ICLR 2022 Submitted_

### Official Review · Reviewer_unQi · 2021-11-02

**Correctness:** 4
**Technical Novelty And Significance:** 2
**Empirical Novelty And Significance:** Not applicable
**Recommendation:** 3
**Confidence:** 3

**Main Review:**

## Strengths
1. The problem of cyclic causality is an interesting problem.
2. The idea of **sufficient representation** is interesting and can be further developed.

## Weaknesses
1. Unfortunately, this paper is rushed/unfinished and needs a significant amount of work to reach the quality of a conference paper.
2. The paper has many grammatical mistakes, including misspelling the word *STRUCTURAL* in the title.
3. The theoretical advances in the paper are not enough for a conference paper. The authors need to show that they can solve a real-world problem using the proposed Meta-SCM. It is unclear if tying the definition to data will make the definition flexible or not.
4. Continuing #4, there are no experiments to validate the proposed definitions.
5. The paper is full of unscholarly language and hyperbole. Here are several examples from the introduction:
    * there are still **some important** theoretical problems
    * They have **many convenient** properties
    * But there is a **strong need** to go beyond acyclic SCMs.
    * However, they are still not well understood.
    * We propose a **totally novel** dimension


**Summary Of The Paper:**

This paper proposes an extension of SCMs that allows modeling cycles in the causality. The main contribution of the paper is providing definitions for the meta-SCM via the concept of active mechanisms and effectively connecting the data to mechanisms.

**Summary Of The Review:**

While the proposed idea of meta-SCM is thought-worthy, the current shape of the paper is far below the quality bars of scholarly conferences such as ICLR. The authors need to add empirical validation of the proposed idea and overhaul the writing of the paper. They also need to show that their proposed meta-SCM has enough novel methodology contributions.

---

> ### Author Response · Authors · 2021-11-21
> **Preliminary knowledge of structural causal models with cycles are required to evaluate our paper**
>
> Thanks anyway for your time and effort. Respectfully, we have to disagree with much of  the criticisms strongly.
>
> > Unfortunately, this paper is rushed/unfinished and needs a significant amount of work to reach the quality of a conference paper.
>
> Our paper addresses the hard theoretical problem of modeling cyclic causality with the meta-SCM framework, which is a mutation of the SCM framework mainly differs on how to connect data to mechanisms. Our framework is a novel theoretical contribution, which can be regarded as a sophisticated integration of SCM in causal inference, active set method in mathematics, information accounts of causality in philosophy and SMS assumption in representation learning.
>
> > The paper has many grammatical mistakes, including misspelling the word STRUCTURAL in the title.
>
> We have corrected some grammatical mistakes and misspelling in our updated pdf.
>
> > The theoretical advances in the paper are not enough for a conference paper. The authors need to show that they can solve a real-world problem using the proposed Meta-SCM. It is unclear if tying the definition to data will make the definition flexible or not.  ... there are no experiments to validate the proposed definitions.
>
> Respectfully, we strongly disagree with the statements that are rather unsubstantiated.
>  It's actually very clear that our meta-SCM nested the SCM as special case of setting $A_k=\mathcal{I}$ for each sample $k$, which is a flexible framework.
>
> > The paper is full of unscholarly language and hyperbole. Here are several examples from the introduction:
> > 1. there are still some important theoretical problems
> > 2. They have many convenient properties
> > 3. But there is a strong need to go beyond acyclic SCMs.
> > 4. However, they are still not well understood.
> > 5. We propose a totally novel dimension
>
> The reviewer lists a number of sentences as examples of "unscholarly language and hyperbole" and we agree to not use unscholarly language and hyperbole. Respectfully, we however have to say that this is just a subjective evaluation by a non-expert on the research problem of cyclic causality. For example, the second sentence "They have many convenient properties" is more of commonsense among researchers. In fact, the paper -- "Foundations of structural causal models with cycles and latent variables."which is published on on Ann. Statist. recently, presents the same sentence "They have many convenient properties" at the beginning of the second paragraph. Henceforth, we tend to think the reviewer is not familiar with the paper[1], which is the SOTA work on theoretical aspects of modeling cyclic causality. And that paper is done by a top team for more than 5 years of hardworking, which provides valuable preliminary knowledges to catch up the recent developments and can help to understand the contribution of our submission. Thus, no offense, we believe the reviewer might be able to assess our paper more objectively after reading that paper.
>
> [1] Stephan Bongers. Patrick Forré. Jonas Peters. Joris M. Mooij. "Foundations of structural causal models with cycles and latent variables." Ann. Statist. 49 (5) 2885 - 2915, October 2021.
>
> **Conclusive comment from the authors to the review:** We feel that the assessment by this review is rather subjectively and we believe the reviewer is a non-expert on this particular research problem of modeling cyclic causality who consider common sense as  "unscholarly language and hyperbole".

---

> > ### Comment · Reviewer_unQi · 2021-11-27
> > **Response**
> >
> > Thank you for your response. As a reviewer, my evaluation is that this paper necessarily needs experiments to show the effectiveness of the ideas.
> >
> > Also, I agree with all four of Reviewer uRUL initial points. This paper is not ready for publication in the current form.

---

> > > ### Author Response · Authors · 2021-11-28
> > > **All four of Reviewer uRUL initial points are mostly based on misunderstanding**
> > >
> > > Thank you for your response.  As we have addressed, all four of Reviewer uRUL initial points are mostly based on misunderstanding of paper. And Reviewer uRUL's further concerns on "how the proposed model can be useful for a real problem" have also been addressed by a toy example of personalization incentives in online short-video platform.
> > >
> > > Respectfully, we hope the this reviewer can offer us constructive suggestions which is an integrated part of review process.

---

### Official Review · Reviewer_mwX7 · 2021-11-03

**Correctness:** 4
**Technical Novelty And Significance:** 4
**Empirical Novelty And Significance:** 4
**Recommendation:** 8
**Confidence:** 4

**Main Review:**

This was a very interesting paper to read, and quite different than the typical conference submission. The authors address one of the most fundamental problems in causal modeling: sufficient representation. I found the model proposed to be both intuitive and general enough to address many concerns such as feedback, equilibrating systems, etc. My largest concern is in practicality: from reading it is not clear to me how one would either learn or parameterize these models. It's not clear how one would recover these indices or distinguish samples between indices, perform interventional calculus, or parameterize the models to make interventional queries. I would be interested to hear the authors' thoughts, though I do realize that these questions are out of scope for the current submission. In short, this paper is a refreshing though provoking piece of work that I believe could be the basis for many interesting future directions.

**Summary Of The Paper:**

This paper presents a new lens on causal graphical models from a lens of fuller generality. Rather than considering models under assumptions such as acyclicity or other constraints that enable tractable modeling, the authors consider the notion of a meta causal model which, when indexed, includes SCM as one possible snapshot. The authors give two additional assumptions (1) sparse mechanism shift and (2) sufficient activated mechanisms, and describe the implications.

**Summary Of The Review:**

Interesting, though provoking paper that proposes a novel paradigm for causal modeling in general systems that are often unrepresentable in current causal graphical frameworks.

---

> ### Author Response · Authors · 2021-11-21
> **We thank the reviewer for his/her appreciation of the paper.**
>
> Thank you for your approval of our work, we are very pleased that you enjoyed it!
>
> Our proposed meta-SCM framework increases its representation capability through a novel dimension on how to connect data to mechanisms. The way that how one learn or parameterize these models should depend on the specific domain of interest and assumptions on active mechanisms. We currently don't have a well-developed answer to your concerns, but we believe efforts on developing causal calculus are needed and deep probabilistic programming tools might help to address the challenges.
> The questions your raised are both important and interesting and will be further explored in future other submissions.

---

### Official Review · Reviewer_uRUL · 2021-11-09

**Correctness:** 2
**Technical Novelty And Significance:** 2
**Empirical Novelty And Significance:** 1
**Recommendation:** 3
**Confidence:** 4

**Main Review:**

**Strengths**

1. Introducing a new way to formulate changing causal mechanisms using sample-wise active sets. The authors have claimed its stronger expressiveness power in modeling unsolvable SCM and cyclic relations than using auxiliary context variables.

2. Covering a broad spectrum of prior work which handles feedback loops between observed variables.

**Weaknesses**

The motivation of this paper is not clear. In the abstract & intro, the goal was originally to solve cyclic relations but how the proposed meta-SCM and SAM in the main content can be used to tackle the problem was not explored. This paper introduced merely a conceptual framework instead of a working system and contained a few unsupported claims which I feel not confident to justify their correctness. The advantages of the proposed active set approach over the classic way to model nonstationarity using context variables are not clear. How the proposed SAM can be coded as a training architecture is questionable.

1. *Motivation is not clear*: What is the central problem being addressed in this paper? This paper seemed to tackle cyclic causal relations in the beginning but then switched to the modeling of sample-wise causal mechanisms through active sets of edges.

2. *No theoretical guarantee or experiment*: This paper introduced merely a conceptual framework instead of a working system.  Many critical claims in the paper are not supported by proofs or experiment results, such as the proposed *Sufficient Activated Mechanisms (SAM)* can recover causal structure if only how many factors have been activated in the sample are known. Questions such as how the proposed meta-SCM is to be built, how the SAM prior can be coded as regularization, under what conditions the proposed framework can work, have not been explored.

3.  *Effectiveness is limited*: I'm not convinced that this is an effective approach. Representing nonstationarity in data using active sets of causal edges might have some advantages over one-hot encoded context variables, such as the dimensionality might be smaller if there exist large numbers of nonstationarity regimes. However, the active sets are also limited in the sense that they cannot model changing causal relations where only the causal strengths vary, for example, for modeling a spring-mass system where spring stiffness wears out over time. Most importantly, I do not think manually encoding the discrete context variables with active sets of edges is a novel idea to the causal community.

4. *Lack of justification for SAM*: as a counterexample (starfish) to argue against sparse mechanism shift (SMS), the authors stated that "changing the output information of the centre node (instead itself) can lead to shift on all functional relationships with its child nodes, which can affect almost all factors simultaneously." I'd like to say this is actually something expected by SMS, as the intervention only changes the marginal distribution of the center node while all conditional relations remain stationary, which still results in the changing of distributions of all factors simultaneously. Also, I can't think of a way to encode the proposed SAM into a learning rule.

**Summary Of The Paper:**

This paper proposes to use **active set** of causal edges, instead of auxiliary context variable or domain index to characterize nonstationary causal relations in data. In addition, the authors propose Sufficient Activated Mechanisms (SAM) to replace Sparse Mechanism Shift (SMS) as the inductive bias (assumption) for causal discovery and inference.

Developing a principled way to characterize mechanism shifts in causal settings is definitely an interesting idea and potentially publishable. However, this paper introduced merely a conceptual framework instead of a working system with theoretical guarantees and contained a few unsupported claims which I feel not confident to justify their correctness. I strongly recommend the authors code the proposed meta-SCM and SAM in a training architecture and validate their correctness in both theoretical and empirical perspectives and submit the to future venues.

**Summary Of The Review:**

I have to reject this paper in the current version. The proposed idea is interesting but without supporting proofs or empirical experiments, it is very hard for ICLR community to justify its correctness and judge its applicability to real-world problems.

---

> ### Author Response · Authors · 2021-11-21
> **The reviewer's misunderstanding for our paper.**
>
> Thank you for your detailed review. We hope to clarify with our comments in what we believe to be misconceptions.
>
> > This paper proposes to use **active set of causal edges**, instead of auxiliary context variable or domain index to characterize nonstationary causal relations in data. In addition, the authors propose Sufficient Activated Mechanisms (SAM) to **replace** Sparse Mechanism Shift (SMS) as the inductive bias (assumption) for causal discovery and inference.
>
> *Response:* We propose the active set of causal mechanisms of a given sample $k$, denoted as $A_k \subseteq \mathcal{I}$.  The number of causal edges might be much greater than $|\mathcal{I}| $, which  implies that the **"active set of causal edges" is not**  a concept proposed in our paper.  Moreover, we **do not** suggest using SAM to "replace" SMS, instead, we claim that  "SAM hypothesis defines a subclass of meta-SCMs of interest for performing causal inferences and reasoning, consistent with the role of the ICM Principle/SMS hypothesis in the SCM framework." In short,  SAM is a novel causal principle for causal inference which is closely related to SMS assumption. Henceforth, we feel the reviewer **does not correctly understand our paper**.
>
> > Developing a principled way to characterize mechanism shifts in causal settings is definitely an interesting idea and potentially publishable. However, this paper introduced merely a conceptual framework instead of a working system with theoretical guarantees and contained a few unsupported claims which I feel not confident to justify their correctness. I strongly recommend the authors code the proposed meta-SCM and SAM in a training architecture and validate their correctness in both theoretical and empirical perspectives and submit the to future venues.
>
> *Reponse:* We should acknowledge that there is **a significant theoretical contribution**  and acknowledge the contribution correctly. We propose a meta structural causal model framework, through adding the novel dimension on connecting data to mechanisms described by active set method, which can be used to circumvent technique complications in cyclic SCMs. Moreover, a novel SAM hypothesis is proposed within the meta-SCM framework for causal inference and learning.
> Since causal modeling with cycles research is pretty at its early stage in which there is no existing widely used benchmarks, it is our future work of interest to code our propose framework into a working system into an architecture and validate it in a proper setting.  Indeed, many papers are accepted at ICLR which do not have contain experiments; this is not a weakness of the paper.
> The reviewers expressed their concerns in detail later, and we will deal with them in detail accordingly.
>
> There are many aspects of causality research, including "Epistemology", "Metaphysics", "Methodology", "Semantics" (See in Illari & Russo (2014)). Sincerely, we then hope the reviewer try to assess our contribution with a more open mind. Respectfully, we recommend to read the book Illari & Russo (2014) which includes friendly introduction of many critical ideas in this paper, e.g.,  level of abstraction.
>
> [1] Phyllis Illari and Federica Russo. Causality: Philosophical theory meets scientific practice. OUP
> Oxford, 2014.

---

> > ### Author Response · Authors · 2021-11-21
> > **Detailed response to the weaknesses**
> >
> > 1. Motivation is not clear: What is the central problem being addressed in this paper? This paper seemed to tackle cyclic causal relations in the beginning but then switched to the modeling of sample-wise causal mechanisms through **active sets of edges**.
> > *Response:* First,  we propose the active set of causal mechanisms **rather than the active sets of edges**. Second, our motivation is to develop a framework which is able to modeling systems with cyclic causal relations. We have shown that our meta-SCM framework, obtained by adding a dimension of connecting data to mechanisms to SCM framework with the active set method,  is able to  circumvent technical complications when cycles present. And further, the SAM hypothesis corresponding to the role of SMS for SCM is proposed within our novel framework.
> >
> > 2. No theoretical guarantee or experiment: This paper introduced merely a conceptual framework instead of a working system. Many critical claims in the paper are not supported by proofs or experiment results, such as the proposed Sufficient Activated Mechanisms (SAM) can recover causal structure if only how many factors have been activated in the sample are known. Questions such as how the proposed meta-SCM is to be built, how the SAM prior can be coded as regularization, under what conditions the proposed framework can work, have not been explored.
> > *Response:* Our meta-SCM framework is able to model systems with cyclic causality, which is guaranteed by Theorem 7 with a formal proof(p.4). It is our future work of interest to code our proposed framework into a working system and validate it in a proper setting given its early stage status on research in modeling causal cycles.
> >
> > 3. Effectiveness is limited: I'm not convinced that this is an effective approach. Representing nonstationarity in data using active sets of causal edges might have some advantages over one-hot encoded context variables, such as the dimensionality might be smaller if there exist large numbers of nonstationarity regimes. However, the active sets are also limited in the sense that they cannot model changing causal relations where only the causal strengths vary, for example, for modeling a spring-mass system where spring stiffness wears out over time. Most importantly, I do not think manually encoding the discrete context variables with active sets of edges is a novel idea to the causal community.
> > *Response:* We do not propose "active sets of causal edges". It is important to clarify that we are **not manually encoding the discrete context variables with active sets of edges**. First, the mechanism for an individual can be activated/inactivated by randomness in the same context, which suggests the two concepts are different. Second, It's true that both context variable and active set method are used to describe nonstationarity, however, "context" is more of a concept at the population level for description, while "active mechanism" is essentially an individual level concept. Floridi emphasizes that we can get into conceptual muddles very quickly by failing to notice two different levels of abstraction(LoA), while in fact the Simpson’s paradox can be seen as a misspecification of LoA [1].
> >
> > 4. Lack of justification for SAM: as a counterexample (starfish) to argue against sparse mechanism shift (SMS), the authors stated that "changing the output information of the centre node (instead itself) can lead to shift on all functional relationships with its child nodes, which can affect almost all factors simultaneously." I'd like to say this is actually something expected by SMS, as the intervention only changes the marginal distribution of the center node while all conditional relations remain stationary, which still results in the changing of distributions of all factors simultaneously. Also, I can't think of a way to encode the proposed SAM into a learning rule.
> > *Response:* It is an interpretation depends on how you formally define a causal mechanism. If you consider the structural equation as the formal definition of causal mechanism, then our claim is supported by the form of intervened structural equations. The SMS has been a useful principle for causal learning(e.g. [2]), we mention the potential  applications of SAM through discussing the connections between them.
> >
> > [1] Phyllis Illari and Federica Russo. Causality: Philosophical theory meets scientific practice. OUP Oxford, 2014.
> >
> > [2] Francesco Locatello, Ben Poole, Gunnar R\'atsch, Bernhard Sch\'olkopf, Olivier Bachem, and Michael
> > Tschannen. Weakly-supervised disentanglement without compromises. In International Confer-
> > ence on Machine Learning, pp. 6348–6359. PMLR, 2020.
> >
> > **Conclusive comment from the authors to the review:** We thank the reviewer, again, for the review. We clarify with our comments in what we believe to be misconceptions.

---

> > > ### Comment · Reviewer_uRUL · 2021-11-24
> > > **Further clarification for sample-wise active mechansims**
> > >
> > > Thanks for the clarifications. What actually confused me a lot when reading the paper is how the proposed model can be useful for a real problem. Here are my concerns.
> > >
> > > 1. How can one know which causal mechanisms are active for each sample? In real-world applications, one may know this segment/domain of data samples have different causal mechanisms from another segment of data, so one can denote the differences by adding a context variable node in the graph, for instance, segment index $u=1, 2, 3, ...$. I think knowing beforehand the active mechanisms of each sample is a too strong assumption and thus not very effective in practice. Can the authors clarify how one can use it even in a toy example?
> > >
> > > 2. How can one use the proposed Sufficient Activated Mechanism (SAM) as a prior to reject a plausible (but wrong) hypothesis? In particular, by assuming "every causal mechanism is sufficiently often activated across samples in the dataset", how does the hypothesis test, or estimation procedures get additional power to reject the wrong hypothesis of the underlying mechanisms? How this principle can be used is not as straightforward as how Sparse Mechanism Shift (SMS) can be encoded as a sparsity constraint.
> > >
> > > 3. Furthermore, the statement that "the proposed Sufficient Activated Mechanisms (SAM) can recover causal structure if only how many factors have been activated in the sample are known" is without explanation or proof. Also, how can one know how many factors (I guess you mean conditional independent mechanisms?) are active in each sample?

---

> > > > ### Author Response · Authors · 2021-11-26
> > > > **A toy example in the real world for our proposed framework**
> > > >
> > > > Though our paper manifests its theoretical and conceptual novelty, your concerns how the proposed model can be useful for a real problem are of interest by all reviewers, which is not directly answered by this particular submission. Here we give a toy example in the real world.
> > > >
> > > > Offering incentives (e.g., coupons at Amazon, discounts at Uber and video bonuses at Tiktok) to users is a strategy widely used by online platforms to increase user engagement and platform revenue. Specifically, the Tiktok(抖音) in China offers personalized cash equivalent bonuses to increase user engagement, and the bonus for the engagement activity of a given user(e.g. watching videos) is obtained only if the bonus pendant is activated (the bonus pendant can be assumed to be activated randomly). As a toy example here, the context for all users are the same, but differ in whether their bonus pendant are activated or not. Thus, as a direct response to **Q1**, the active mechanism is somehow a better description than the context segment index. Moreover, for **Q2&Q3**, there are sufficiently many users activate the bonus pendant suggested by our novel assumption, which is at least not a direct interpretation of the SMS.
> > > >
> > > > Thanks for your valuable questions very much, and the toy example has already addressed them to some extent. Indeed, many papers are accepted at ICLR which do not contain experiments; this is not a weakness of the paper. Our paper has proposed a novel approach to address an important theoretical problem in causal modeling and reveals its grace by connecting it to existing results.

---

> > > > > ### Comment · Reviewer_uRUL · 2021-11-29
> > > > > **Thanks for the response but I keep my score**
> > > > >
> > > > > Dear Authors of Paper 963,
> > > > >
> > > > > I thank the authors for the response. However, unfortunately, since this is not a paper to propose unsolved research questions, but rather a paper to provide concrete solutions, I think more needs to be done to explain under what (sufficient) conditions/assumptions the proposed problem setup can be solved (with proof), such as the answer I've been keeping asking for, i.e., why "the proposed Sufficient Activated Mechanisms (SAM) can recover causal structure if only how many factors have been activated in the sample are known."
> > > > >
> > > > > Also, I strongly agree with Reviewer unQi that for a complete theoretical investigation, numerical experiments are necessary to validate its effectiveness. I retain my original decision for these reasons.
> > > > >
> > > > > Best regards,
> > > > >
> > > > > Reviewer uRUL

---

> > > > > > ### Author Response · Authors · 2021-11-30
> > > > > > **Thanks for your review, but another misconception**
> > > > > >
> > > > > > Thanks for your time and review, but we have to clarity the there is another **misconception** with respect.
> > > > > >
> > > > > > You are keep asking: "the proposed Sufficient Activated Mechanisms (SAM) can recover causal structure **if only** how many factors have been activated in the sample are known." Here is the corresponding sentence in our paper:
> > > > > > > In Example 2, the unsolvable cyclic SCM .... On the contrary, in the meta-SCM framework under SAM assumption, we are able to learn a causal factorization Eq. (2), even when only knowing how many factors have activated, but not which ones.
> > > > > >
> > > > > > The correct interpretation should be:
> > > > > > 1) we are able to learn a causal factorization Eq. (2);
> > > > > > 2) In a weaker case that **only knowing how many factors have activated, but not which ones**,  we still might be able to learn a causal factorization Eq. (2).
> > > > > >
> > > > > > It is definitely not our intention to ignore your question deliberately, but just we thought it is quite obvious. We are quite disappoint to your decision, given that we have clarified **many misconceptions** and encouraged you to assess our paper with evaluation dimension other than experiments.  Science advances by discovery, our paper addresses the fundamental problem in causality research with a novel framework which **brings more than enough new information to causality research.**
> > > > > >
> > > > > > Though we are not satisfied with your score, but **sincerely** thanks for your time and review which inspires us to make the paper easier to understand. We are not satisfied and ask:What if we try much harder to clarify the Reviewer uRUL's misconceptions? What if the Reviewer unQi had not been a non-expert on cyclic causality research who consider common sense as "unscholarly language and hyperbole"?  Then would we be able to get better scores? We don't know the answers,  but only hope for the received scores reflect the true value of our paper objectively as possible.

---

### Author Response · Authors · 2021-12-02
**Summary of our responses**

Dear AC and reviewers:

In summary, Reviewer uRUL and Reviewer unQi insist their scores mainly because of the lack of experiments to validate its effectiveness.

> The Reviewer uRUL: ..., numerical experiments are necessary to validate its effectiveness. I retain my original decision for these reasons.

> The Reviewer unQi: ... that this paper necessarily needs experiments to show the effectiveness of the ideas.

Henceforth, a simple question is "**what kind of experiments on cyclic causality modeling do you expect to be effective enough? Would you please recommend us any published paper to be effective enough?**". As a paper on cyclic causality modeling, we definitely unsatisfied with the rejections only because of the lack of experiments.  Experiments are surely important that we will explore in the future, but it would be rather ridiculous to reject a paper on **certain research problems**, e.g. string theory, mainly because of the lack of experiments.

Our proposed framework addresses an important theoretical problem in causal modeling, associated with a novel causal principle, which is the only causal inference framework explicitly explores **the novel dimension of connecting data to mechanisms** to the best of knowledge.

---

> ### Comment · Reviewer_unQi · 2021-12-02
> **Response**
>
> Dear AC,
>
> The authors have called me a "non-expert" three times in their responses in a derogatory sense. This language clearly violates [ICLR's code of conduct on "bullying and harassment"](https://iclr.cc/public/CodeOfConduct). I will refrain from further commenting on this paper. The reviewers do not deserve this language.
>
> @Author: Please ask your questions from your "expert friends".

---

> > ### Author Response · Authors · 2021-12-03
> > **Many reviewers just admit they are non-expert for a research problem in the OpenReview Platform**
> >
> > Dear Reviewer unQi,
> >
> > **Sincerely thanks** for you time and review, though which is not constructive enough. We have to **clarify our statement.**
> >
> > We **only** evidently say "you are a non-expert **in a particular research problem**",  which is true for everyone on earth, but still you might be able to be very constructive. We definitely don't expect you might feel being offended by a statement such as "a non-expert in a particular research problem", **given the fact many reviewers just admit they are non-expert for a research problem in the OpenReview Platform.**
> >
> > The following sentence is what we have said:
> > >  Respectfully, we however have to say that this is **just a subjective evaluation by a non-expert on the research problem of cyclic causality.** For example, the second sentence "They have many convenient properties" is more of commonsense among researchers. ....
> >
> > > We feel that the assessment by this review is rather subjectively and we believe the reviewer is **a non-expert on this particular research problem of modeling cyclic causality** who consider common sense as "unscholarly language and hyperbole".
> >
> > The questions of "what kind of experiments on cyclic causality modeling do you expect to be effective enough? Would you please recommend us any published paper to be effective enough?" are **constructive feedbacks**.  In fact, it is not appropriate to reject a paper in this research problem mainly because of the lack of experiments.

---

### Decision · Program_Chairs · 2022-01-20

**Decision:**

Reject

**Comment:**

This paper proposes a meta-structural causal model framework, to increase the representation capability of structural equation models. It also considers how to connect data to mechanisms. The paper is conceptually interesting. However, on the technical side, reviewers feel that without supporting proofs or empirical experiments, it is hard to justify the correctness of the proposal and judge its applicability to real-world problems.

As authors claimed in their response, "it is our future work of interest to code our proposed framework into a working system and validate it in a proper setting given its early stage status on research in modeling causal cycles." I think some future version of the paper might be a great contribution to the field if a working system were included.